# Sleep-Disordered Breathing Risk with Comorbid Insomnia Is Associated with Mild Cognitive Impairment

**Christian Agudelo [1,2,]\***, **Alberto R. Ramos [1,2]**, **Xiaoyan Sun [1,2]**, **Sonya Kaur [1,2]**, **Dylan F. Del Papa [1]**, **Josefina M. Kather [1]**, **Douglas M. Wallace [1]** and on behalf of the Alzheimer's Disease Neuroimaging Initiative (ADNI) [†]

1 Department of Neurology, University of Miami Miller School of Medicine, Miami, FL 33136, USA; a.ramos1@med.miami.edu (A.R.R.); xsun@med.miami.edu (X.S.); ssk109@med.miami.edu (S.K.); dylan.delpapa@jhsmiami.org (D.F.D.P.); jkather@med.miami.edu (J.M.K.); dwallace@med.miami.edu (D.M.W.)

2 Evelyn McKnight Brain Institute, University of Miami Miller School of Medicine, Miami, FL 33136, USA

\* Correspondence: cxa427@med.miami.edu

† Data used in preparation of this article were obtained from the Alzheimer's Disease Neuroimaging Initiative (ADNI) database (adni.loni.usc.edu). As such, the investigators within the ADNI contributed to the design and implementation of ADNI and/or provided data but did not participate in analysis or writing of this report. A complete listing of ADNI investigators can be found at: http://adni.loni.usc.edu/wp-content/uploads/how_to_apply/ADNI_Acknowledgement_List.pdf.

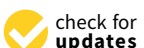

**Featured Application: This study aims to identify candidate modifiable risk factors of cognitive disease. Future applications of this study include the optimization of sleep to ameliorate cognitive decline.**

**Abstract:** Introduction: Few studies have evaluated the combined association between SDB with comorbid insomnia and mild cognitive impairment (MCI). To test the hypothesis that SDB with comorbid insomnia is associated with greater odds of MCI than either sleep disorder independently, we used ADNI data to evaluate cross-sectional associations between SDB risk with comorbid insomnia status and MCI. Methods: Participants with normal cognition or MCI were included. Insomnia was defined by self-report. SDB risk was assessed by modified STOP-BANG. Logistic regression models evaluated associations between four sleep disorder subgroups (low risk for SDB alone, low risk for SDB with insomnia, high risk for SDB alone, and high risk for SDB with insomnia) and MCI. Models adjusted for age, sex, BMI, APOE4 genotype, race, ethnicity, education, marital status, hypertension, cardiovascular disease, stroke, alcohol abuse, and smoking. Results: The sample (*n* = 1391) had a mean age of 73.5 ± 7.0 years, 44.9% were female, 72.0% were at low risk for SDB alone, 13.8% at low risk for SDB with insomnia, 10.1% at high risk for SDB alone, and 4.1% at high risk for SDB with insomnia. Only high risk for SDB with comorbid insomnia was associated with higher odds of MCI (OR 3.22, 95% CI 1.57–6.60). Conclusion: Studies are needed to evaluate SDB with comorbid insomnia as a modifiable risk factor for MCI.

**Keywords:** insomnia; sleep-disordered breathing; sleep apnea; cognitive impairment; dementia

## 1. Introduction

Over 50 million people globally have cognitive impairment and dementia. The global economic burden of cognitive disease exceeded USD 1 trillion in 2018 [1]. Individual sleep disturbances, including sleep-disordered breathing (SDB) and insomnia, have each been associated with dementia [2]. Combined, SDB with comorbid insomnia has been associated with lower quality of life, more daytime sleepiness, and greater mood disturbances than either insomnia alone or SDB alone [3]. However, few studies have explored the combined association between SDB with comorbid insomnia and cognitive impairment,

which, together, may convey more severe cognitive consequences. In a meta-analysis of 18 prospective studies examining sleep disturbances (i.e., insomnia, SDB, and other sleep problems) and the risk of dementia [2], only three studies assessed both SDB and insomnia, yet none of their analyses accounted for SDB with comorbid insomnia [4–6].

In the general adult population, the prevalence of insomnia is 10–40% [7,8], while the prevalence of SDB ranges between 9 and 38% [9]. Estimates of the prevalence of SDB with comorbid insomnia vary depending on the population studied. Among adults presenting to sleep clinics with SDB symptoms, 39–58% also report insomnia symptoms. In contrast, among those presenting with insomnia complaints, 29–67% have at least mild SDB [10]. That is, SDB with comorbid insomnia occurs together more frequently than expected based on prevalence estimates of each sleep disorder alone.

SDB with comorbid insomnia has been hypothesized as a bidirectionally related entity [11]. Obstructive respiratory events occurring during the initial wake-to-sleep transition may contribute to difficulties with sleep onset, while worsening sleep-disordered breathing during REM sleep may trigger premature early awakenings. One study showed that 90% of awakenings among participants with insomnia were preceded by an obstructive respiratory event, although participants attributed the awakenings to psychological (stress, ruminations, anxiety) or physiological (nocturia, discomfort, temperature) causes [12]. Baseline SDB has also been shown to predict incident insomnia symptoms at 7.5-year follow-up in a longitudinal observational study (OR = 1.83; 95% CI = 0.94–3.54; $p$ = 0.074) [13]. Though treating SDB with positive airway pressure (PAP) therapy has been shown to improve some insomnia complaints (i.e., middle insomnia), it can fail to improve others (early and late insomnia) [14–19]. Conversely, a smaller body of research has demonstrated that insomnia may be a risk factor for SDB. Experimentally induced sleep fragmentation, sleep restriction, and sleep deprivation (all features of insomnia) have been shown to exacerbate respiratory disturbances during sleep among those with and without SDB [20–26]. Treatment of insomnia with cognitive behavioral therapy for insomnia in patients with SDB with comorbid insomnia resulted in a reduction in SDB severity [27]. With growing evidence of the bidirectional influence between SDB and insomnia, it is possible that SDB with comorbid insomnia may uniquely contribute to the risk of mild cognitive impairment (MCI). To our knowledge, studies examining the relationship between SDB or insomnia and cognitive impairment have rarely controlled or examined the concurrent influence of the other prevalent sleep disorder.

Thus, the current study stratified participants with normal cognition and MCI by risk for SDB and insomnia status to explore the associations of each of these two sleep disorders and their combined occurrence with MCI. We hypothesize that participants with SDB with comorbid insomnia have greater odds of MCI than participants without either sleep disorder. We further hypothesize that participants with comorbid insomnia and high risk for SDB have greater odds of MCI than participants with insomnia alone or SDB risk alone.

## 2. Materials and Methods

### 2.1. The ADNI Database and Participants

Data used in the preparation of this manuscript were obtained from the Alzheimer's Disease Neuroimaging Initiative (ADNI) database (adni.loni.usc.edu). The ADNI was launched in 2003 as a public-private partnership, led by Principal Investigator Michael W. Weiner, M.D. The primary goal of ADNI has been to test whether serial neuroimaging, other biological markers, and clinical and neuropsychological assessment can be combined to measure the progression of mild cognitive impairment (MCI) and early Alzheimer's disease (AD). For up-to-date information, see www.adni-info.org.ata.

ADNI participants were enrolled from 86 clinical sites in the United States and Canada. At baseline visits, participants were between 50 and 90 years of age, and their cognitive function ranged from normal cognition to Alzheimer's dementia. This analysis excluded those participants with dementia at baseline. Baseline anthropometric, sociodemographic, and clinical data were utilized for this analysis and are described in the Main Confounders

subsection. Detailed descriptions of sleep variables (i.e., SDB risk and insomnia status) and cognitive outcomes are described below. The variable names for all the data used within the ADNI database are in Table S1 (Supplementary Materials).

## 2.2. Primary Outcome: Normal Cognition vs. MCI

The primary outcome was the presence of normal cognition or MCI. The ADNI published criteria for subject classification elsewhere [28] and abide by established clinical criteria for the diagnosis of Alzheimer's disease [29]. Participants with normal cognition and MCI scored between 24 and 30 in the Mini-Mental State Examination (MMSE). Participants with MCI had a subjective memory complaint, objective memory loss measured by education-adjusted scores on the Wechsler Memory Scale Logical Memory III, a global Clinical Dementia Rating (CDR) score of 0.5, absence of significant levels of impairment in other cognitive domains, preserved activities of daily living, and an absence of dementia.

## 2.3. Insomnia

Insomnia was assessed by self-report. Participants were asked about the presence or absence of "insomnia" in item 22 of the baseline Diagnosis and Symptoms Checklist, which was completed by the participant and their study partner.

## 2.4. Risk for SDB

The risk for SDB was assessed using a modified STOP-BANG. The STOP-BANG is a validated questionnaire that stratifies the risk of SDB by probing eight characteristics: loud snoring, daytime sleepiness/tiredness, observed choking or gasping during sleep, hypertension, BMI greater than 30 mg/kg$^2$, age, neck circumference, and sex [30]. The ADNI did not systematically assess snoring, observed choking/gasping in sleep, or neck circumference; thus, those characteristics were excluded from the modified STOP-BANG. As a proxy for the measure of daytime sleepiness, we used item eight of the baseline Diagnosis and Symptoms Checklist, which asks whether "low energy" is present or absent. A modified STOP-BANG score was constructed with that proxy measures for daytime sleepiness, along with history of hypertension, objective BMI, age, and self-reported sex. High risk for SDB was defined by a score of 4 or higher. The positive predictive value of a STOP-BANG of 4 or higher for SDB in prior studies was 95% in a sleep clinic population and 77% in a surgical patient population [31].

## 2.5. Main Confounders

Statistical analyses adjusted for the following baseline demographic covariates: age, self-reported sex (male or female), objective BMI (kg/m$^2$), APOE4 genotype (0, 1, or 2 alleles), race (White, Black or African American, Asian, American Indian or Alaska Native, or Native Hawaiian or Other Pacific Islander), ethnicity (not Hispanic or Latino, Hispanic or Latino, or unknown), education (years), marital status (married, never married, divorced, widowed, or unknown). Additionally, confounders included a self-reported history of hypertension, cardiovascular disease, stroke, alcohol abuse, and smoking.

## 2.6. Analytical Sample

Our analytical sample used baseline visit data from three ADNI study phases: ADNI-1, ADNI-GO, and ADNI-2. Data were collected between 2005 and 2013. Data were downloaded from the ADNI website on 9 February 2021. The initial analytical sample consisted of all participants with baseline cognitive data, of which there were 1740. Participants with dementia were excluded (339), leaving 1401 participants. All 1401 participants had a reported insomnia status and data for full SDB risk assessment using the modified STOP-BANG. An additional 10 participants were excluded due to insufficient data for covariate adjustment, each missing APOE4 data. The final analytical sample consisted of 1391 participants (Figure 1).

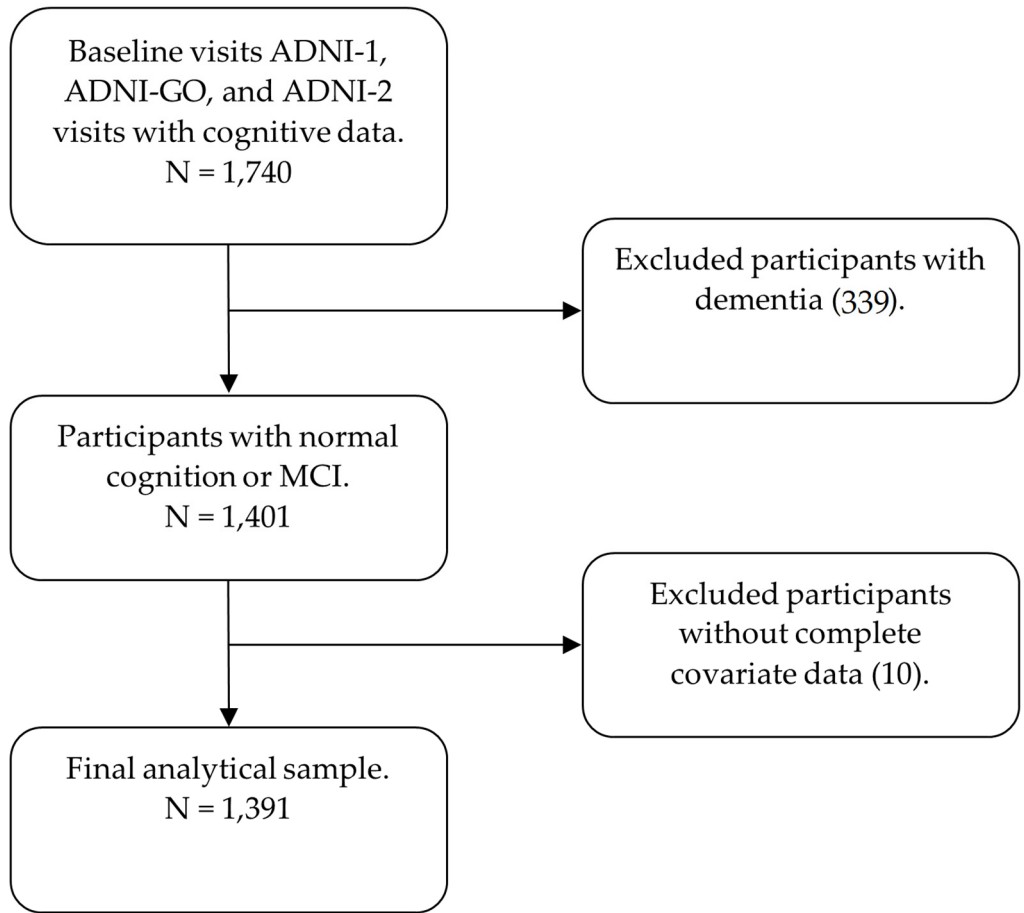

**Figure 1.** Diagram depicting the analytical sample.

*2.7. Statistical Analyses*

First, four sleep disorder categorical subgroups were created: (1) low risk for SDB alone, (2) low risk for SDB with insomnia, (3) high risk for SDB alone, and (4) high risk for SDB with insomnia. Descriptive statistics were generated to compare each group by demographic variables (Table 1). Differences in group means and proportions among sleep disorder groups were examined using ANOVA and chi-squared tests, respectively. Post hoc pairwise analyses were used to assess the differences in group means and proportions. Bonferroni correction was used to account for multiple pairwise comparisons.

Logistic regression models were constructed to examine the associations between the four sleep disorder categorical subgroups and the odds of normal cognition or MCI. Initially, low risk for SDB alone was the reference. Model 1 adjusted for demographic, anthropometric, and genetic variables: age, sex, BMI, and APOE4 allele number. Model 2 further adjusted for social variables: race, ethnicity, education, and marital status. Model 3 further adjusted for clinical and behavioral variables: hypertension, cardiovascular disease, stroke, alcohol abuse, and smoking status.

To further examine the comparisons between sleep disorder groups with MCI, logistic regression models were repeated with high risk for SDB with comorbid insomnia as the reference exposure. The same confounders as previously described were included.

For all analyses, $p < 0.05$ was defined as statistically significant. Statistical analyses were performed with SPSS Version 27 (IBM, Armonk, NY).

**Table 1.** Demographic characteristics of the study population per sleep disorder subgroup.

| | Sleep Disorder Groups | | | | |
|---|---|---|---|---|---|
| | Low Risk for SDB Alone [A] | Low Risk for SDB with Insomnia [B] | High Risk for SDB Alone [C] | High Risk for SDB with Insomnia [D] | Study Sample |
| Size, *n* (% study sample) | 1002 (72.0) | 191 (13.7) | 141 (10.1) | 57 (4.1) | 1391 (100) |
| Age, mean years (SD) | 73.5 (6.9) | 72.8 (7.5) | 74.0 (6.9) | 74.1 (7.4) | 73.5 (7.0) |
| Sex, female % | **47.7** [C D] | **62.3** [A C D] | **7.1** | **28.1** [C] | **44.8** |
| BMI, mean mg/kg$^2$ (SD) | **26.2 (4.5)** | **26.8 (5.2)** | **31.0 (4.8)** [A B] | **31.8 (5.9)** [A B] | **27.0 (5.0)** |
| APOE4 alleles | | | | | |
| 0, % | 56.5 | 62.8 | 61.0 | 56.1 | 57.8 |
| 1, % | 35.3 | 31.4 | 30.5 | 42.1 | 34.6 |
| 2, % | 8.2 | 5.8 | 8.5 | 1.8 | 7.6 |
| Race | | | | | |
| White, % | 92.9 | 92.7 | 91.5 | 86.0 | 92.5 |
| Black or African American, % | 4.3 | 4.2 | 3.5 | 10.5 | 4.5 |
| Asian, % | 1.6 | 2.1 | 0.7 | 1.8 | 1.6 |
| American Indian or Alaska Native, % | 0.2 | 0.0 | 0.7 | 0.0 | 0.2 |
| Native Hawaiian or Other Pacific Islander, % | 0.2 | 0.0 | 0.0 | 0.0 | 0.1 |
| More than one, % | **0.6** | **0.5** | **3.5** [A] | **1.8** | **0.9** |
| Unknown, % | 0.2 | 0.5 | 0.0 | 0.0 | 0.2 |
| Ethnicity | | | | | |
| Non-Hispanic or Latino, % | 96.0 | 95.8 | 96.5 | 100.0 | 96.2 |
| Hispanic or Latino, % | 3.3 | 4.2 | 2.8 | 0.0 | 3.2 |
| Unknown, % | 0.7 | 0.0 | 0.7 | 0.0 | 0.6 |
| Education, mean years (SD) | 16.0 (2.8) | 16.3 (2.6) | 16.3 (3.0) | 15.9 (2.7) | 16.1 (2.8) |
| Marital status | | | | | |
| Married, % | **73.2** | **68.6** | **85.8** [A B D] | **64.9** | **73.5** |
| Never married, % | **3.7** | **7.9** [C] | **0.7** | **5.3** | **4.0** |
| Divorced, % | 10.8 | 7.9 | 5.0 | 8.8 | 9.7 |
| Widowed, % | 12.1 | 14.7 | 7.8 | 19.3 | 12.3 |
| Unknown, % | 0.3 | 1.0 | 0.7 | 1.8 | 0.5 |
| Hypertension, % | **40.4** | **44.0** | **95.7** [A B] | **86.0** [A B] | **48.4** |
| Stroke, % | **1.0** | **0.5** | **3.5** [A] | **0.0** | **1.2** |
| Cardiovascular disease, % | **61.7** | **69.6** | **95.7** [A B] | **93.0** [A B] | **67.5** |
| Alcohol abuse, % | **4.1** | **1.0** | **5.0** | **8.8** [B] | **4.0** |
| Smoking, % | 39.0 | 35.6 | 44.0 | 54.4 | 39.7 |

Differences in group means and proportions among sleep disorder groups were examined using ANOVA and chi-squared tests, respectively. Post hoc pairwise analyses were used to assess the differences in group means and proportions. Rows with statistically significant differences ($p < 0.05$) are in bold. The statistically larger proportion or mean is identified by the superscripted letter(s) (A–D) of the group(s) with the smaller proportion(s) or mean(s).

## 3. Results

Descriptive statistics of the sample and comparisons among the four sleep disorder groups are presented in Table 1. The sample had a mean age of 73.5 ± 7.0 years, was 44.8% female, had a mean BMI of 27 ± 5 mg/kg$^2$, 57.8% lacked APOE4 alleles, 92.5% were white, 96.2% were non-Hispanic, and 62.7% had MCI. Among the sample, 72.0% were at low risk for SDB alone, 13.7% were at low risk for SDB with insomnia, 10.1% were at high risk for SDB alone, and 4.1% were at high risk for SDB with insomnia.

Several significant differences existed between the four stratified sleep disorder groups (Table 1). Regardless of insomnia status, participants at high risk for SDB were more often male, had greater BMI, and more often reported a history of hypertension or cardiovascular

disease. Participants at high risk for SDB without insomnia more often had a history of stroke than those at low risk for SDB without insomnia. Additionally, participants at high risk for SDB with insomnia reported a history of alcohol abuse more often than those at low risk for SDB with insomnia.

In logistic regression models of sleep disorder subgroups and MCI, with low risk for SDB alone as the reference group, only high risk for SDB with insomnia was significantly associated with greater odds (OR 3.22, 95% CI 1.57–6.60) of MCI after full covariate adjustment (Table 2). Neither high risk for SDB alone nor low risk for SDB with insomnia was significantly associated with greater odds of MCI relative to SDB alone. Furthermore, in logistic regression models with high risk for SDB with insomnia as the reference, low risk for SDB alone (0.31, 95% CI 0.15–0.64), low risk for SDB with insomnia (0.31, 95% CI 0.14–0.66), and high risk for SDB alone (0.43, 95% CI 0.20–0.95) were each significantly associated with lower odds of MCI after full covariate adjustment (Table 3).

**Table 2.** Associations between sleep disorder groups and MCI with low risk for SDB alone as reference.

|  | Mild Cognitive Impairment | | |
|---|---|---|---|
|  | Model 1 OR (95% CI) | Model 2 OR (95% CI) | Model 3 OR (95% CI) |
| Low risk for SDB with insomnia | 0.95 (0.69–1.32) | 0.99 (0.71–1.38) | 0.98 (0.70–1.38) |
| High risk for SDB alone | 1.43 (0.94–2.18) | 1.42 (0.92–2.18) | 1.40 (0.89–2.21) |
| High risk for SDB with insomnia | 2.88 ** (1.43–5.8) | 3.14 ** (1.55–6.36) | 3.22 *** (1.57–6.60) |

The results of logistic regression used to evaluate the associations of sleep disorder groups (low risk for SDB with insomnia, high risk for SDB alone, and high risk for SDB with insomnia, each relative to low risk for SDB alone) with the odds of MCI (versus normal cognition). Model 1 adjusted for sociodemographic (age and sex), anthropometric (BMI), and genetic (APOE4 allele number) variables. Model 2 further adjusted for social variables (race, ethnicity, education, and marital status). Model 3 further adjusted for clinical (hypertension, cardiovascular disease, and stroke) and behavioral (alcohol abuse and smoking) variables. ** $p < 0.01$; *** $p < 0.001$.

**Table 3.** Associations between sleep disorder groups and MCI with high risk for SDB with insomnia as reference.

|  | Mild Cognitive Impairment | | |
|---|---|---|---|
|  | Model 1 OR (95% CI) | Model 2 OR (95% CI) | Model 3 OR (95% CI) |
| Low risk for SDB alone | 0.35 ** (0.17–0.70) | 0.32 ** (0.16–0.65) | 0.31 ** (0.15–0.64) |
| Low risk for SDB with insomnia | 0.33 ** (0.16–0.7) | 0.32 ** (0.15–0.67) | 0.31 ** (0.14–0.66) |
| High risk for SDB alone | 0.5 (0.23–1.07) | 0.45 * (0.21–0.98) | 0.43 * (0.20–0.95) |

The results of logistic regression used to evaluate the association of sleep disorder groups (low risk for SDB alone, low risk for SDB with insomnia, and high risk for SDB alone, each relative to high risk for SDB with insomnia) with the odds of MCI (versus normal cognition). Model 1 adjusted for sociodemographic (age and sex), anthropometric (BMI), and genetic (APOE4 allele number) variables. Model 2 further adjusted for social variables (race, ethnicity, education, and marital status). Model 3 further adjusted for clinical (hypertension, cardiovascular disease, and stroke) and behavioral (alcohol abuse and smoking) variables. * $p < 0.05$; ** $p < 0.01$.

## 4. Discussion

In this cross-sectional analysis of baseline ADNI data, among older adults with normal cognition and MCI, high risk for SDB with comorbid insomnia was associated with greater odds of MCI when compared to each other sleep disorder group. Neither low risk for SDB with insomnia nor high risk for SDB alone was significantly associated with greater odds of MCI when compared to low risk for SDB alone.

Existing research in the relationship between SDB and cognitive impairment has rarely considered the concurrent influence of insomnia. Similarly, few studies concerning the relationship between insomnia and cognitive impairment have examined the comorbidity of SDB. Our study highlights the importance of examining these two common sleep

disorders concurrently. In prior studies of SDB that did not adjust for comorbid insomnia, SDB was prospectively associated with incident cognitive decline at 4-year follow-up among community-dwelling older women [32], and with incident Alzheimer's dementia (RR = 1.20, 95% CI: 1.03–1.41) and vascular dementia (RR = 1.23, 95% CI: 1.04–1.46) in a meta-analysis of three studies [2]. In studies of insomnia lacking adjustment for comorbid SDB, insomnia predicted all-cause dementia at 8-year follow-up in a retrospective cohort study of older adults [4], at 3-year follow-up in a case–control study of middle-aged adults [33], and in a meta-analysis of five population-based prospective cohort studies (RR = 1.53 95% CI: 1.07–2.18) [34], while chronic insomnia predicted incident all-cause dementia at 4-year follow-up in a longitudinal observational study [35].

Studies have also independently associated SDB and insomnia with lower function in certain neurocognitive domains. SDB has been associated with deficits in attention, vigilance, and information processing speed [36]. Furthermore, sleep fragmentation in SDB has been associated with deficits in sustained attention and reaction time and visuospatial deficits, while hypoxia in SDB has been associated with deficits in sustained attention and reaction time [37]. In contrast, insomnia among older adults has been inconsistently associated with deficits among various neurocognitive domains [38]. Nevertheless, treatment of insomnia has been associated with improved complex vigilance task function and reduced simple vigilance task function [39]. These studies, though, have not adjusted for the comorbid presence of SDB and insomnia. Further research should examine the association between SDB with comorbid insomnia and neurocognitive function domains.

Two recent analyses of cognitively normal adults examined sleep disturbances and neurocognitive function and did account for concurrent SDB and insomnia. An observational study of middle-aged Hispanic/Latino adults reported that insomnia severity was not predictive of 7-year neurocognitive decline neither after adjustment for SDB severity nor after excluding moderate-to-severe SDB in sensitivity analyses [40]. Another recent analysis observed that adults with SDB and comorbid insomnia had lower neurocognitive function than previously published age-normative values [41]. Importantly, neither study examined the association between concurrent sleep disorders and cognitive impairment.

In our modeling of the association between sleep disorder subgroups and MCI, only those at high risk for SDB with comorbid insomnia had greater odds of MCI. Insomnia and SDB could be independent processes, often occurring comorbidly, each independently contributing to the risk of MCI in our study population. However, neither low risk for SDB with insomnia nor high risk for SDB alone were associated with MCI in our study. A bidirectional relationship may exist between SDB and comorbid insomnia that imparts unique contributions to the risk of cognitive impairment. Concurrent insomnia and SDB could be associated with distinct alterations in sleep physiology. Sleep microstructure [42–46], arousal thresholds [42,47], and sleep fragmentation [10] are aspects of sleep physiology affected in both SDB and insomnia, any of which could mediate a relationship between SDB with comorbid insomnia and the risk of cognitive impairment. Sleep fragmentation has been prospectively associated with an increased risk of incident Alzheimer's disease at 6-year follow-up [48]. Sleep fragmentation has also been associated with neuroinflammation [49] and oxidative stress [50], each of which is implicated in Alzheimer's dementia. Similarly, aspects of sleep microstructure have been associated with Alzheimer's disease, including slower occipital electroencephalogram activity during rapid eye movement sleep [51], poorly formed sleep spindles [51], and diminished slow-wave activity in sleep [52]. Some aspects of sleep microstructure have been associated with cerebral beta-amyloid, a hallmark of Alzheimer's disease, including diminished slow-wave activity [53–56] and diminished slow oscillations [53]. Other aspects of sleep microstructure have been associated with cerebral tau burden, another hallmark of Alzheimer's dementia, including the decoupling of slow oscillations from spindles [57]. Further research is needed to understand the effect of SDB with comorbid insomnia on sleep physiology and whether sleep physiology mediates a unique association between SDB with comorbid insomnia

and cognitive impairment. Conversely, further research is needed to explore the potential effects that cognitive impairment may have on SDB with comorbid insomnia.

*Limitations*

This study has limitations inherent to secondary analyses of cross-sectional epidemiological data. Causality cannot be inferred from cross-sectional analyses. Our use of a modified STOP-BANG only provides a measure of SDB risk and not the gold standard diagnostic measure of SDB provided by polysomnography. Nevertheless, the use of a modified STOP-BANG allowed for the systematic assessment of SDB risk of each participant in a study without a priori measures of SDB. Similarly, our use of the self-reported presence or absence of insomnia is a limited measure of insomnia and does not substitute for a gold standard clinical diagnostic interview. The ADNI is not a population-based sample. By design, people with MCI and dementia were oversampled, so results may not generalize to the community-dwelling population.

## 5. Conclusions

These data suggest that older individuals with SDB and comorbid insomnia may be a group at high risk for cognitive impairment. Further studies that utilize validated measures of insomnia and SDB are needed to better characterize cross-sectional and longitudinal associations between SDB with comorbid insomnia and cognitive disease. Additional studies are needed to identify the mechanisms of SDB with comorbid insomnia that mediate its relationship with cognitive impairment, as these mechanisms may serve as targets of intervention to mitigate cognitive decline.

**Supplementary Materials:** The following are available online at https://www.mdpi.com/article/10.3390/app12052414/s1, Table S1: List of ADNI database variables used in analyses.

**Author Contributions:** Conceptualization, C.A., A.R.R., X.S., S.K., J.M.K. and D.M.W.; methodology, C.A., A.R.R., X.S., S.K. and D.M.W.; formal analysis, C.A., A.R.R., X.S., S.K. and D.M.W.; data curation, C.A. and X.S.; software, C.A.; writing—original draft preparation, C.A. and D.F.D.P.; writing—review and editing, C.A., A.R.R., X.S., S.K., D.F.D.P., J.M.K. and D.M.W.; visualization, C.A.; supervision, C.A., A.R.R., X.S. and D.M.W.; project administration, C.A. and D.M.W. All authors have read and agreed to the published version of the manuscript.

**Funding:** This research was supported by the Evelyn F. McKnight Brain Research Foundation through the Evelyn F. McKnight Neurocognitive Fellowship for Christian Agudelo.

**Institutional Review Board Statement:** The study was conducted according to the guidelines of the Declaration of Helsinki and approved by the local Institutional Review Boards of each participating ADNI site. A complete list of participating ADNI sites and IRB details can be obtained at adni.loni.usc.edu.

**Informed Consent Statement:** Informed consent was obtained from all subjects or authorized representative involved in the study. Details of informed consent procedures can be obtained at adni.loni.usc.edu.

**Data Availability Statement:** The data used in this study are available from the Alzheimer's Disease Neuroimaging Initiative study database (adni.lni.usc.edu).

**Acknowledgments:** Institutional support was provided by the McKnight Brain Institute at the University of Miami Miller School of Medicine. Data collection and sharing for this project were funded by the Alzheimer's Disease Neuroimaging Initiative (ADNI) (National Institutes of Health Grant U01 AG024904) and DOD ADNI (Department of Defense award number W81XWH-12-2-0012). ADNI is funded by the National Institute on Aging, the National Institute of Biomedical Imaging and Bioengineering, and through generous contributions from the following: AbbVie; Alzheimer's Association; Alzheimer's Drug Discovery Foundation; Araclon Biotech; BioClinica, Inc.; Biogen; Bristol-Myers Squibb Company; CereSpir, Inc.; Cogstate; Eisai Inc.; Elan Pharmaceuticals, Inc.; Eli Lilly and Company; EuroImmun; F. Hoffmann-La Roche Ltd. and its affiliated company Genentech, Inc.; Fujirebio; GE Healthcare; IXICO Ltd.; Janssen Alzheimer Immunotherapy Research & Development, LLC.;

Johnson & Johnson Pharmaceutical Research & Development LLC.; Lumosity; Lundbeck; Merck & Co., Inc.; Meso Scale Diagnostics, LLC.; NeuroRx Research; Neurotrack Technologies; Novartis Pharmaceuticals Corporation; Pfizer Inc.; Piramal Imaging; Servier; Takeda Pharmaceutical Company; and Transition Therapeutics. The Canadian Institutes of Health Research is providing funds to support ADNI clinical sites in Canada. Private sector contributions are facilitated by the Foundation for the National Institutes of Health (www.fnih.org). The grantee organization is the Northern California Institute for Research and Education, and the study is coordinated by the Alzheimer's Therapeutic Research Institute at the University of Southern California. ADNI data are disseminated by the Laboratory for Neuro Imaging at the University of Southern California.

**Conflicts of Interest:** The authors declare no conflict of interest. The funders had no role in the design of the study; in the collection, analyses, or interpretation of data; in the writing of the manuscript; or in the decision to publish the results.

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
