# Peer review of "Sleep-Disordered Breathing Risk with Comorbid Insomnia Is Associated with Mild Cognitive Impairment"

_applsci, doi:10.3390/app12052414_

Round 1

Reviewer 1 Report

In general, this study is interesting. Some minor suggestions/changes:

*Abstract: Better description of the study aim, and the conclusion must be related to the hypothesis

* Introduction is ok. The description of "MCI" abbreviation (line 78) must be before in line 73.

* Methods. Improve the description of the statistical analyzes performed

* Conclusion: It is ok. Limitations well described.

Author Response

February 7, 2022

RE: Submission of revised manuscript “applsci-1462157”

Dear Reviewer,

Thank you for your thorough review, comments, and the opportunity to improve our manuscript titled “Sleep Disordered Breathing Risk with Comorbid Insomnia is Associated with Mild Cognitive Impairment.”

We are grateful for your feedback and observations for how to improve the manuscript. Your insights provide us the opportunity to better communicate the scientific efforts of the authors, investigators, and participants of the Alzheimer’s Disease Neuroimaging Initiative. Below is an itemized response to each of your major points.  We have carefully reviewed your comments and revised accordingly. We detailed our responses below each comment. We have attached a copy of the manuscript resubmission with tracked changes (as a Word document).

Kind Regards,

Christian Agudelo, MD
Clinical Instructor
Evenly F. McKnight Neurocognitive Scholar
Division of Sleep Medicine
Department of Neurology
University of Miami Miller School of Medicine

Reviewer Comments and Responses

Reviewer 1

1. Abstract: Better description of the study aim, and the conclusion must be related to the hypothesis

Response: Thank you for this observation. We agree that the abstract would benefit if the study’s central hypothesis was included. The introduction section of the abstract has been edited to explicitly state the central hypothesis of the study. The primary result of the study, as stated in the results section of the abstract, now directly answers the primary aim and hypothesis of the study. These changes are in lines 18-21 and are highlighted as tracked changes in the attached resubmission Word document.

2. Introduction is ok. The description of "MCI" abbreviation (line 78) must be before in line 73.

Response: Thank you for identifying this error. The abbreviation for mild cognitive impairment, MCI, is now presented at its first occurrence. In the attached manuscript resubmission, this occurs in lines 85-86.

3. Methods. Improve the description of the statistical analyzes performed

Response: We agree that the manuscript lacked sufficient description of the statistical analyses used. Drafts prior to submission included a final subsection in the methods section: “2.7 Statistical Analysis.” By some error, this section was not included in the submitted manuscript. We have included it in the attached resubmission (lines 162-181).

Reviewer 2 Report

Authors propose the use of logistic regression to study the correlation between mild cognitive impairment and sleep disorder. 

Whilst the article is well detailed from the medical point of view, it lacks an explanation about why logistic regression, among several statistical and machine learning methods has been used. Do the authors assume that the relationships are not linear?

Authors must briefly describe logistic regression and discuss why it has been used.

Moreover, values in Tables 2 and 3 must be described and discussed in the text. What do these values represent?

Author Response

February 7, 2022

RE: Submission of revised manuscript “applsci-1462157”

Dear Reviewer,

Thank you for your thorough review, comments, and the opportunity to improve our manuscript titled “Sleep Disordered Breathing Risk with Comorbid Insomnia is Associated with Mild Cognitive Impairment.”

We are grateful for your feedback and observations for how to improve the manuscript. Your insights provide us the opportunity to better communicate the scientific efforts of the authors, investigators, and participants of the Alzheimer’s Disease Neuroimaging Initiative. Below is an itemized response to each of your major points.  We have carefully reviewed your comments and revised accordingly. We detailed our responses below each comment. We have attached a copy of the manuscript resubmission with tracked changes (as a Word document).

Kind Regards,

Christian Agudelo, MD
Clinical Instructor
Evenly F. McKnight Neurocognitive Scholar
Division of Sleep Medicine
Department of Neurology
University of Miami Miller School of Medicine

Reviewer Comments and Responses

Reviewer 2:

  1. Authors propose the use of logistic regression to study the correlation between mild cognitive impairment and sleep disorder. Whilst the article is well detailed from the medical point of view, it lacks an explanation about why logistic regression, among several statistical and machine learning methods has been used. Do the authors assume that the relationships are not linear? Authors must briefly describe logistic regression and discuss why it has been used.

Response: We agree that the manuscript lacked sufficient description of the statistical analyses used. Drafts prior to submission included a final subsection in the methods section: “2.7 Statistical Analysis.” By some error, this section was not included in the submitted manuscript. We have included it in the attached resubmission (lines 162-181). We specifically used logistic regression because this allowed for comparison of categorical predictors/exposures (the four sleep disorder subgroups) with two categorical outcomes (normal cognition or mild cognitive impairment). Logistic regression evaluated the relative odds that each exposure/predictor group had for each outcome. We did not consider using linear regression analysis to compare four categorical predictors/exposures and two categorical outcomes as the results of such an analysis, when possible, provide results that are difficult to interpret clinically.

  1. Moreover, values in Tables 2 and 3 must be described and discussed in the text. What do these values represent?

Response: To better describe the data presented in Tables 2 and 3 within the results section, we have edited the last paragraph of the results section (199-206). Specifically, the most relevant odds ratios (and associated 95% confidence intervals) presented in Tables 2 and 3 are now included within the text of the results section. These changes are highlighted in the attached resubmission Word document as tracked changes.
